# Preoperative versus Postoperative Compensation of the Contralateral Normal Kidney in Patients Treated with Radical Nephrectomy for Renal Cell Carcinoma

**DOI:** 10.3390/jcm10214918

**Published:** 2021-10-24

**Authors:** Chung-Un Lee, Hyunsoo Ryoo, Jae-Hoon Chung, Wan Song, Minyong Kang, Hyun-Hwan Sung, Byong-Chang Jeong, Seong-Il Seo, Seong-Soo Jeon, Hyun-Moo Lee, Hwang-Gyun Jeon

**Affiliations:** 1Department of Urology, Samsung Medical Center, Sungkyunkwan University School of Medicine, Seoul 06351, Korea; iatronices@naver.com (C.-U.L.); dr.jhchung@gmail.com (J.-H.C.); wan.song@samsung.com (W.S.); dr.minyong.kang@gmail.com (M.K.); hyunhwan.sung@samsung.com (H.-H.S.); bc2.jung@samsung.com (B.-C.J.); seongil.seo@samsung.com (S.-I.S.); seongsoo.jeon@samsung.com (S.-S.J.); hyunmoo.lee@samsung.com (H.-M.L.); 2Department of Urology, VHS Medical Center, Seoul 05368, Korea; rhs860918@hanmail.net

**Keywords:** renal compensation, radical nephrectomy, renal cell carcinoma

## Abstract

Background: We sought to identify the factors affecting renal compensatory processes that occur preoperatively as well as postoperatively in patients treated with radical nephrectomy (RNx) for renal cell carcinoma (RCC). Methods: We retrospectively reviewed the records of 906 patients treated with RNx for RCC. We defined the early compensatory process (process 1) as compensatory adaptation of the contralateral normal kidney (CNK) before RNx. We defined the late compensatory process (process 2) as compensatory adaptation of the CNK after RNx. Total compensation was defined as the combination of these two processes. Multivariable logistic regression analyses were used to identify significant factors associated with processes 1, 2 and total compensation. Results: Mean preoperative, 1-week, and 5-year postoperative estimated glomerular filtration rates (eGFR) were 84.5, 57.6 and 63.7 mL/min/1.73 m^2^, respectively. Female sex (*p* < 0.001), lower body mass index (BMI) (*p* < 0.001), absence of hypertension (*p* = 0.019), lower preoperative eGFR (*p* < 0.001), larger tumor volume (*p* < 0.001), and larger CNK volume (*p* < 0.001) were significantly associated with process 1. Younger age (*p* = 0.019), higher BMI (*p* < 0.001), and absence of diabetes mellitus (DM) (*p* = 0.033) were significantly associated with process 2. Female sex (*p* < 0.001), younger age (*p* < 0.001), absence of DM (*p* = 0.002), lower preoperative eGFR (*p* < 0.001), and larger tumor (*p* = 0.001) and CNK volumes (*p* < 0.001) were significantly associated with total compensation. Conclusions: Different factors affected each compensatory process. Process 1 made a greater contribution to the entire renal compensatory process than process 2.

## 1. Introduction

Kidney cancer accounts for 3% of all adult malignancies, and renal cell carcinoma (RCC) is the most common primary malignancy of the kidney [1,2]. Standard treatment options for patients with RCC include radical nephrectomy (RNx) and partial nephrectomy (PNx), which are frequently performed. Although several studies have reported that PNx is associated with superior renal function outcomes [3,4,5,6,7] in addition to cardiovascular and overall survival benefits [3,5,8], RNx remains a common surgical option for RCC. Whenever RNx is selected for RCC treatment, postoperative renal function must be taken into consideration.

Following RNx, the contralateral normal kidney (CNK) enlarges and increases its function. The compensatory adaptive mechanisms involved in sensing the absence of the tumor-side kidney and growth of the contralateral kidney are incompletely understood [9]. However, several studies have identified factors that lead to changes in estimated glomerular filtration rate (eGFR) after nephrectomy [1,3,10,11,12,13,14,15,16,17,18,19,20]. Moreover, many studies have also investigated factors that can affect renal compensation [11,21,22,23,24,25,26,27]. Preoperative factors, including tumor size, can affect compensation by the CNK, suggesting that compensation by the CNK occurs before RNx, in addition to after this procedure.

However, in previous studies, compensatory adaptation was not subdivided into distinct pre- and postoperative processes [10,21,22,23,24,25,26,27], and in most studies, only pre- and postoperative GFR have been compared at last follow-up [10,22,26]. Here, we divided the entire renal compensatory process into a preoperative process and a postoperative process and investigated factors that affect each compensatory process.

## 2. Materials and Methods

### 2.1. Ethnics Approval

This retrospective study was approved by the Institutional Review Board of Samsung Medical Center (IRB No. 2018-11-079), and the IRB waived the requirement for informed consent due to the retrospective nature of this study. All study protocols were performed in accordance with the principles of the Declaration of Helsinki.

### 2.2. Study Population and Variables Included in the Study

A total of 906 patients underwent RNx for RCC between October 1996 and December 2013. Clinical data on patient sex, age, body mass index (BMI), diabetes mellitus (DM) status, presence of hypertension (HTN), and laboratory data were obtained from individual patient medical records collected at the time of hospital admission for surgery. Preoperative computed tomography (CT) was performed within 3 months before surgery. Serum creatinine level was measured preoperatively, and postoperatively at 1 week (measured several times during hospitalization until postoperative 1 week; the highest value was selected), 3 months, and 1, 3, 5 and 10 years. GFR was estimated using the abbreviated Modification of the Diet and Renal Disease (MDRD) equation: eGFR (mL/min/1.73 m^2^) = 186.3 × creatinine − 1.154 × age − 0.203 (×0.742 if female) [25].

### 2.3. Volume Measurement

CT was performed with a multi-detector CT scanner (GE Medical Systems, Milwaukee, WI, USA) using a standard abdominopelvic imaging protocol. CT images of the renal parenchyma with a 5-mm slice thickness were obtained. Venous scans of the entire abdomen were obtained with a 60-s delay after contrast administration. CT images of the venous phase were exported to the Xelis software (Infinitt, Seoul, Korea). Detailed volume measurements were performed using the same procedures as described in our previous studies [21,23,24]. We chose a threshold of >50 HU, and the observer manually rendered the tumor area; the software calculated the 3D tumor volume, tumor-side real kidney volume, and remnant kidney volume [21,23,24].

### 2.4. Evaluation for Process of Renal Compensation

We reasoned that CNK begins compensating as an adaptive process before RNx when a tumor exists and that this compensation persists after RNx. Based on this assumption, we defined the early compensatory process (process 1) as the compensatory adaptation of the CNK before RNx, and we defined the late compensatory process (process 2) as the compensatory adaptation of the CNK after RNx. Total compensation was defined as the combination of these two processes. Compensation was evaluated by comparing changes in eGFR after various time points. Process 1 was evaluated by comparing eGFR 1-week postoperatively to the preoperative eGFR, while process 2 was evaluated by comparing the postoperative follow-up eGFR to the 1-week postoperative eGFR. Total compensation was evaluated by comparing the postoperative follow-up eGFR to the preoperative eGFR. Process 1 was considered successful if patients had a high 1-week postoperative eGFR/preoperative eGFR ratio. Specifically, the top 25% of patients with a high 1-week postoperative eGFR/preoperative eGFR ratio (B/A in Figure 1) were deemed to have achieved success with respect to process 1. Likewise, the top 25% of patients with a high 5-year postoperative eGFR/1-week postoperative eGFR ratio (F/B in Figure 1) and 5-year postoperative eGFR/preoperative eGFR ratio (F/A in Figure 1) were deemed to have achieved success with respect to process 2 and total compensation, respectively. Subsequently, we identified factors affecting success in process 1, process 2, and total compensation.

### 2.5. Statistical Analysis

Continuous variables are summarized as means and medians (interquartile range, IQR) while categorical variables are summarized as absolute values and percentages. Multivariable logistic regression analyses were used to identify factors significantly associated with the success of process 1, process 2, and total compensation. All tests were two-tailed, and *p* < 0.05 was considered statistically significant. All statistical analyses were performed using IBM SPSS Statistics, version 21.0 (IBM Co., Armonk, NY, USA).

## 3. Results

### 3.1. Patient Baseline Characteristics

Table 1 shows the clinical characteristics of the 906 patients who underwent RNx. The number of male and female patients was 631 (69.6%) and 275 (30.4%), respectively. Median patient age was 55 years (IQR, 48–64 years), and median BMI was 24.6 kg/m^2^ (IQR, 22.6–26.6). Median preoperative creatinine level was 0.93 mg/dL (IQR, 0.79–1.08), and median preoperative GFR was 85.0 mL/min/1.73 m^2^ (IQR, 71.4–96.5). Median tumor volume, tumor-side real kidney volume, and CNK volume were 50.7 mL (IQR, 23.4–105.8), 170.1 mL (IQR, 144.9–195.6), and 177.3 mL (IQR, 154.3–202.8), respectively.

### 3.2. Changes in Mean GFR in Patients Treated with RNx

Figure 1 shows the changes in mean eGFR before and after RNx in patients with RCC. The mean preoperative eGFR was 84.5 mL/min/1.73 m^2^. The mean 1-week postoperative eGFR was reduced by 31.8% (57.6 mL/min/1.73 m^2^) in comparison to the mean preoperative value. The mean eGFR increased gradually following the operation. The mean 5- and 10-year postoperative eGFR values were 10.5% and 12.8% higher (63.7 mL/min/1.73 m^2^ and 65.0 mL/min/1.73 m^2^), respectively, than the mean 1-week postoperative value, but these values were lower than the mean preoperative eGFR by 24.6% and 23.0%, respectively.

### 3.3. Factors Affecting the Success of Process 1

Table 2 presents the results of the univariable and multivariable logistic regression analyses performed to identify factors associated with renal compensation from the preoperative period to one week postoperatively in patients treated with RNx for RCC (Univariable logistic regression analyses are presented in Appendix A). Multivariable regression analysis showed that renal compensation was positively associated with female sex (*p* < 0.001), lower BMI *(p* < 0.001), absence of HTN (*p* = 0.019), lower preoperative eGFR (*p* < 0.001), larger tumor volume (*p* < 0.001), and larger CNK volume (*p* < 0.001).

### 3.4. Factors Affecting the Success of Process 2

Multivariate analysis of factors associated with renal compensation from 1 week to 5 years postoperatively in patients treated with RNx revealed that younger age (*p* = 0.040), higher BMI (*p* < 0.001), and absence of DM (*p* = 0.019) significantly affected process 2 (Table 3). (Univariable logistic regression analyses are presented in Appendix A).

### 3.5. Factors Affecting the Success of Total Compensation

Among the variables suspected of having an effect on renal compensation from the preoperative period to 5 years postoperatively, female sex (*p* < 0.001), younger age (*p* < 0.001), absence of DM (*p* = 0.002), lower preoperative eGFR (*p* < 0.001), and larger tumor (*p* = 0.001), and CNK volumes (*p* < 0.001) were found to significantly and positively affect renal compensation in the multivariate analysis (Table 4). (Univariable logistic regression analyses are presented in Appendix A).

## 4. Discussion

Compensatory adaptation of the CNK is known to occur in patients treated with RNx for RCC [10,21,22,23,24,25,26,27]. In previous studies on renal compensation, the process of compensatory adaptation has not been divided into specific processes [10,21,22,23,24,25,26,27]. One of the major strengths of this study is that we separated the renal compensatory process into two distinct processes: preoperative and postoperative. To accurately evaluate preoperative compensation, changes in the size of the CNK should ideally be tracked through CT as the tumor grows. However, it is almost impossible to identify compensatory hypertrophy in the CNK during the CT follow-up even if a patient is diagnosed with RCC. A series of previous studies demonstrated that the size of the tumor affects compensation by the CNK. We hypothesized that the tumor influences compensation of the CNK based on these studies. Therefore, we defined process 1 as the compensation that takes place from the time the tumor develops to before the operation. The preoperative process of compensation takes place in both kidneys, while postoperative compensation takes place in the CNK without a tumor. Compensation was evaluated by changes in eGFR before and after RNx. We considered a less dramatic drop in eGFR immediately after RNx to indicate that process 1 was successful. A higher increment in eGFR after surgery over a long follow-up period was considered to indicate the success of process 2. Due to the lack of a universal definition of successful compensation, we arbitrarily defined those patients with a successful process 1 as the top 25% of patients whose eGFR decreased the least immediately after surgery, and those patients with successful process 2 as the top 25% of patients whose overall increment in eGFR was the greatest. In addition, we identified factors affecting the success of process 1, process 2, and total compensation. The division of the entire renal compensation process of RCC patients treated with RNx into processes 1 and 2 allowed us to gain a more comprehensive understanding of renal compensation after RNx.

We found that female sex was positively associated with process 1 and total compensation. If process 1 has a more substantial role than process 2, factors affecting process 1 would affect total compensation to a greater extent than those affecting process 2. Female sex was positively associated with process 1 and total compensation; this indicates that process 1 plays a more important role in total compensation than process 2. A similar conclusion can be drawn from the results presented in Figure 1. Postoperative eGFR showed a sharp decline from preoperative eGFR, but a relatively small difference was observed between 5-year and 1-week postoperative eGFR. Therefore, a slight reduction in eGFR, indicating sufficient compensation during process 1, is important to the entire process of compensatory adaptation of the CNK. 

We showed here that recovery is more likely among those with a lower preoperative eGFR in process 1 and total compensation. In our previous study, we reported that patients with a lower preoperative eGFR had a smaller reduction in postoperative renal function than those with a higher preoperative eGFR [21]. Moreover, other studies have reported similar results [12,17,27]. If preoperative renal function is already reduced, the function of the tumor-side kidney may be poorer than that of the CNK. In such cases, the renal function of the CNK is more important than that of the tumor-side kidney for the total renal function. If patients with lower levels of preoperative eGFR were to undergo RNx, their immediate postoperative eGFR would be less decreased than in those patients with higher preoperative eGFR, because the renal function of the CNK would account for a larger portion of the total renal function.

Compensatory adaptation was positively associated with a larger tumor volume and the CNK volume in process 1 and total compensation. In our previous study, we reported a similar result. Namely, we found that tumor size (>7 cm) was the strongest factor associated with compensatory hypertrophy in the CNK before surgery [24]. This suggests that the CNK volume is larger if compensatory hypertrophy of the CNK has occurred because of a large tumor volume. In addition, immediate postoperative eGFR would likely decrease less if compensation by the CNK is adequate. Tumor volume and function of the CNK therefore share a causal relationship. In addition, the renal function of the tumor-side kidney would be more decreased than that of the CNK if the normal parenchymal volume of the tumor side was due to a lesser extent to the large area of the tumor-side kidney occupied by the tumor. In this scenario, the renal function of the CNK would account for a large portion of the total renal function. Postoperative eGFR would decrease to a lesser extent if a tumor-side kidney with a large tumor was removed than if a tumor-side kidney with a small tumor was removed, because in the former case, the tumor-side kidney would already have a reduced function and the CNK would maintain a relatively higher renal function after RNx. 

Young age was positively associated with process 2 and total compensation. Several previous studies have reported that young age is a preoperative predictor of an increase in eGFR after RNx [1,3,15,16,19,20,22]. Although age was not a significant factor affecting process 1, it was an important factor affecting process 2 and the overall process of renal compensation.

There are some limitations to this study. First, it was retrospective in nature and the results should not be generalized, as analyses were based on data from a single-center tertiary care center. Second, GFR was estimated using the abbreviated MDRD equation. There are some studies indicating that the MDRD equation had limitations for reflecting GFR, and the Chronic Kidney Disease Epidemiology Collaboration (CKD-EPI) equation is better for determining GFR [28,29,30]. Moreover, the Current guideline (2012 KDIGO (Kidney Disease: Improving Global Outcomes) guideline for CKD) suggests to use the CKD-EPI equation [31]. We could not apply the CKD-EPI formula due to the retrospective nature of this study. Third, we arbitrarily defined the success of each process of compensation due to the lack of a universal definition of successful compensation. Our findings may have been different if we had used different definitions of compensatory success. Another limitation of this study was that our approach required measurement of parenchymal volume and eGFR, which could have introduced some variability. Nevertheless, using three-dimensional volume data rather than two-dimensional size data provides more precise information for analyzing the compensatory adaptation of the CNK. A relatively large number of patients and a sufficient follow-up period are strengths of this study.

## 5. Conclusions

Factors significantly affecting each process were different. Changes in the contralateral kidney before RNx, corresponding to process 1, were more important in determining the overall renal compensation than changes in the contralateral kidney after RNx (process 2).

## Figures and Tables

**Figure 1 jcm-10-04918-f001:**
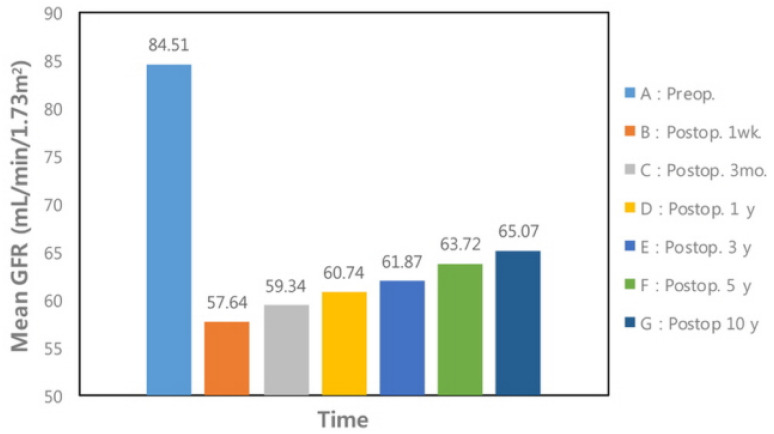
Changes in mean GFR before and after radical nephrectomy in patients with renal cell carcinoma. GFR: Glomerular filtration rate, Postop: Postoperative, Preop: Preoperative.

**Table 1 jcm-10-04918-t001:** Baseline characteristics of patients who underwent radical nephrectomy.

Characteristics	
No. of patients, *n*	906
Sex	
Male, *n* (%)	631 (69.6)
Female, *n* (%)	275 (30.4)
Median age, years (IQR)	55 (48–64)
Median BMI, kg/m^2^ (IQR)	24.6 (22.6–26.6)
Diabetes mellitus, *n* (%)	114 (12.6)
Hypertension, *n* (%)	245 (27.0)
Median creatinine, mg/dL (IQR)	0.93 (0.79–1.08)
Median GFR, mL/min/1.73 m^2^ (IQR)	85.01 (71.45–96.57)
Tumor side	
Right, *n* (%)	423 (46.7)
Left, *n* (%)	483 (53.3)
Median tumor volume, mL (IQR)	50.7 (23.4–105.8)
Median tumor side real kidney volume *, mL (IQR)	170.1 (144.9–195.6)
Median contralateral normal kidney volume, mL (IQR)	177.3 (154.3–202.8)
Median follow-up, month (IQR)	73.5 (52.9–102.9)

* Tumor side real kidney volume was calculated by subtracting the tumor volume from the normally functioning tissue including tumor tissue. GFR, Glomerular filtration rate; IQR, Interquartile range.

**Table 2 jcm-10-04918-t002:** Multivariate logistic regression analysis of factors associated with compensation from preoperative period to postoperative 1 week (process 1) in patients treated with radical nephrectomy for RCC.

	Multivariate	
	HR (95% CI)	*p*
Sex		
Male	Reference	
Female	4.153 (2.645–6.521)	<0.001
BMI	0.866 (0.809–0.926)	<0.001
HTN		
No	Reference	
Yes	0.586 (0.376–0.914)	0.019
Preop. GFR	0.917 (0.904–0.930)	<0.001
Tumor volume	1.003 (1.002–1.015)	<0.001
CNK volume	1.027 (1.020–1.033)	<0.001

HR, Hazard ratio; CI, Confidence interval; BMI, Body mass index; DM, Diabetes mellitus; HTN, Hypertension; Preop., Preoperative; GFR, Glomerular filtration rate; CNK, Contralateral normal kidney.

**Table 3 jcm-10-04918-t003:** Multivariate logistic regression analysis of factors associated with compensation from postoperative 1 week to postoperative 5 years (process 2) in patients treated with radical nephrectomy for RCC.

	Multivariate	
	HR (95% CI)	*p*
Age	0.982 (0.965–0.999)	0.040
BMI	1.120 (1.057–1.187)	<0.001
DM		
No	Reference	
Yes	0.430 (0.212–0.872)	0.019

HR, Hazard ratio; CI, Confidence interval; BMI, Body mass index; DM, Diabetes mellitus; HTN, Hypertension; Preop., Preoperative; GFR, Glomerular filtration rate; CNK, Contralateral normal kidney.

**Table 4 jcm-10-04918-t004:** Multivariate logistic regression analysis of factors associated with compensation from preoperative period to postoperative 5 years (process 1 + 2) in patients treated with radical nephrectomy for RCC.

	Multivariate	
	HR (95% CI)	*p*
Sex		
Male	Reference	
Female	3.036 (1.879–4.905)	<0.001
Age	0.933 (0.912–0.955)	<0.001
DM		
No	Reference	
Yes	0.284 (0.126–0.638)	0.002
Preop. GFR	0.932 (0.908–0.939)	<0.001
Tumor volume	1.003 (1.001–1.005)	0.001
CNK volume	1.019 (1.012–1.026)	<0.001

HR, Hazard ratio; CI, Confidence interval; BMI, Body mass index; DM, Diabetes mellitus; HTN, Hypertension; Preop., Preoperative; GFR, Glomerular filtration rate; CNK, Contralateral normal kidney.

## Data Availability

The dataset used and/or analyzed during the current study is available from the corresponding author upon reasonable request.

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
