# Peer review of "Preoperative versus Postoperative Compensation of the Contralateral Normal Kidney in Patients Treated with Radical Nephrectomy for Renal Cell Carcinoma"

_jcm, 2021, doi:10.3390/jcm10214918_

Round 1
Reviewer 1 Report
Current retrospective study with 906 patients was designed to identify the factors affecting renal compensatory processes occurred preoperatively as well as postoperatively in patients treated with radical nephrectomy (RNx) for renal cell carcinoma (RCC). Total renal compensation was defined as the combination of these two processes.
The study design and presentation are interesting. Result quite expected but anyway important. MDRD eGFR calculation as a limitation should be mentioned because current guidelines (KDIGO CKD guideline 2012) suggest to use CKD-EPI guideline. I can understand that authors have retrospective data but especially MDRD calculation may have calculation deviations in high and low limits. There have been differences in terms of usage of MDRD or CKD-EPI calculations (Grootendorst D et al NDT, 2011, Omuse G et al MBC Nephrology, 2017, Ekrikpo UE et al. PLoS One. 2018;13:e0195443).
Author Response
Thank you for your comment.
Your comment is very important.
We totally agree with you, and we further described about MDRD and CKD-EPI equation in limitation section.
Reviewer 2 Report
Thank to all authors for this valuable study. It is well designed and well concluded. However i have some points that need to be clarify,
- Were all of the patients being followed-up up to 10 years? If not please add this to M&M section.
- In "3.3. Factors affecting the success of process 1." and "3.5. Factors affecting the success of total compensation." sections you found out in multivariate analysis that larger tumor volume is risk factor for process 1 and total compensation. However HRs are 1 in both analyses. Please check that again.
- In general there is no need for the results of univariate analyses, only multivariate analysis is enough. These can be confusing for audience. You can add them as supplementary material.
- First sentence of the Conclusion section seems more suitable for discussion section.
Author Response
Thank you for your constructive and insightful comment.
1. Were all of the patients being followed-up up to 10 years? If not please add this to M&M section.
- We added median follow-up on Table 1, and that explained all patients are not followed-up to 10 years.
2. In "3.3. Factors affecting the success of process 1." and "3.5. Factors affecting the success of total compensation." sections you found out in multivariate analysis that larger tumor volume is risk factor for process 1 and total compensation. However HRs are 1 in both analyses. Please check that again.
- To solve the problems, we represented the number in Table 2, 3, 4 to 3 decimal places. Additionally, we are sorry that there was an error in Table 3. We analyzed all data one more time, and we found that sex is not associated with process 2. We corrected all related matters. We are sure there is no more problems.
3. In general there is no need for the results of univariate analyses, only multivariate analysis is enough. These can be confusing for audience. You can add them as supplementary material.
- We totally agree with you. As your recommendation, we changed the Tables.
4. First sentence of the Conclusion section seems more suitable for discussion section.
- As your recommendation, we moved first sentence of the Conclusion to Discussion section.